# Recurrence Pattern of Cervical Cancer Based on the Platinum Sensitivity Concept: A Multi-Institutional Study from the FRANCOGYN Group

**DOI:** 10.3390/jcm9113646

**Published:** 2020-11-12

**Authors:** Tiphaine de Foucher, Cecile Hennebert, Yohan Dabi, Lobna Ouldamer, Vincent Lavoué, Ludivine Dion, Geoffroy Canlorbe, Pierre Adrien Bolze, François Golfier, Cherif Akladios, Lise Lecointre, Yohan Kerbage, Pierre Collinet, Alexandre Bricou, Xavier Carcopino, Cyrille Huchon, Emilie Raimond, Olivier Graesslin, Clémentine Owen, Cyril Touboul, Marcos Ballester, Emile Darai, Sofiane Bendifallah

**Affiliations:** 1Department of Gynaecology and Obstetrics, Tenon University Hospital, Assistance Publique des Hôpitaux de Paris (AP-HP), 75020 Paris, France; cecile.hennebert@gmail.com (C.H.); clementine.owen@aphp.fr (C.O.); cyril.touboul@gmail.com (C.T.); emile.darai@aphp.fr (E.D.); sofiane.bendifallah@aphp.fr (S.B.); 2Departement of Obstetrics, Gynaecology and Reproductive Medecine, Centre Hospitalier Intercommunal de Créteil, 94000 Créteil, France; yohann.dabi@gmail.com; 3Department of Gynaecology, CHRU de Tours, Hôpital Bretonneau, INSERM unit 1069, 2 boulevard Tonnelé, 37044 Tours, France; l.ouldamer@chu-tours.fr; 4Department of Gynaecology, CHU de Rennes, 35000 Rennes, France; Vincent.LAVOUE@chu-rennes.fr (V.L.); ludivine.dion@chu-rennes.fr (L.D.); 5Assistance Publique des Hôpitaux de Paris (AP-HP), Department of Gynecological and Breast Surgery and Oncology, Pitié-Salpêtrière University Hospital, 75013 Paris, France; geoffroy.canlorbe@aphp.fr; 6Department of Gynaecologic and Oncologic Surgery and Obstetrics, Centre Hospitalier Universitaire Lyon Sud, Hospices Civils de Lyon, Université Lyon 1, 69000 Lyon, France; pierre-adrien.bolze@chu-lyon.fr (P.A.B.); francois.golfier@chu-lyon.fr (F.G.); 7Service de Gynécologie Obstétrique, CHU Hautepierre, 67000 Strasbourg, France; cherif.akladios@gmail.com (C.A.); lise.lecointre@chru-strasbourg.fr (L.L.); 8Department of Gynaecologic surgery, Hôpital Jeanne de Flandre, CHRU LILLE, Rue Eugene avinée, CEDEX, 59037 Lille, France; yohan.kerbage@chru-lille.fr (Y.K.); Pierre.COLLINET@chru-lille.fr (P.C.); 9Department of Obstetrics and Gynaecology, Jean-Verdier University Hospital, Assistance Publique des Hôpitaux de Paris, 93140 Bondy, France; alex.bricou@gmail.com; 10Department of Obstetrics and Gynaecology, Hôpital Nord, APHM, Aix-Marseille University (AMU), Univ Avignon, CNRS, IRD, IMBE UMR 7263, 13397 Marseille, France; xcarco@free.fr; 11Department of Gynaecology, CHI Poissy-St-Germain, Université Versailles-Saint-Quentin en Yvelines, EA 7285 Risques cliniques et sécurité en santé des femmes, Université Versailles-Saint-Quentin en Yvelines, 78000 Versailles, France; cyrillehuchon@yahoo.fr; 12Department of Obstetrics and Gynaecology, Alix de Champagne Institute, Centre Hospitalier Universitaire, 45 rue Cognacq-Jay, 51092 Reims, France; emilie_raimond@hotmail.com (E.R.); olivier.graesslin@gmail.com (O.G.); 13Department of Gynaecologic and Breast Surgery, Groupe Hospitalier Diaconesses Croix Saint Simon, 125 rue d’Avron, 75020 Paris, France; MBallester@hopital-dcss.org

**Keywords:** advanced cervical cancer, recurrence pattern, platinum sensitivity

## Abstract

The standard of care for patients with advanced cervical cancer (ACC) includes platinum-based chemotherapy. The concept of platinum sensitivity is a major prognostic factor for patients with ovarian cancer. The aim of this study was to validate the applicability of the platinum sensitivity concept to ACC patients, and to estimate its prognostic interest in terms of overall survival (OS) and pattern of recurrence (location, timing). Data of women with histologically proven FIGO 2019 stages IB3–IV ACC, treated between May 2000 and November 2017 with platinum-based regimens, were retrospectively abstracted from 12 institutions from the FRANCOGYN Group. Respective 3-year OSs were 52% (95% CI: 40.8%–66.8%), 21.6% (95% CI: 12.6%–37.2%), and 14.6% (95% CI: 4.2%–50.2%), in case of recurrence <6 months, between 6 and 17 months, and ≥18 months (*p* < 0.001). Risk of metastatic or multisite recurrence was significantly higher in case of recurrence <6 months, and risk of local or isolated infradiaphragmatic nodal recurrence was significantly higher in case of recurrence >18 months (*p* < 0.001). In multivariate analysis, platinum sensitivity status was a strong prognostic factor for OS after recurrence, independent of histological grade, lympho-vascular space involvement, final lymph node status, and treatment. Platinum sensitivity status may help to classify patients in three prognostic subgroups for OS after recurrence, and appears to be a strong prognostic factor correlated to the pattern of recurrence.

## 1. Introduction

Cervical cancer (CC) remains the fourth most common cancer worldwide. It is even more common in low- and middle-income countries (LMICs), where it appears to be in second position after breast cancer. With 569,847 new cases and 311,365 deaths worldwide in 2018, CC constitutes a major public health issue [1].

Advanced cervical cancer (ACC) is defined as a tumour of stage IB3–IV according to the International Federation of Gynecology and Obstetrics (FIGO) 2019, and accounts for 40% of all cases on diagnosis [2]. It includes locally advanced CC (LACC—IB3 to IVA) and initially metastatic CC (IVB). Although therapeutic strategies vary according to national and international guidelines, the standard of care is based on first-line concomitant chemoradiotherapy (CCRT) or chemotherapy (CT) alone with cisplatin-based regimens [3,4].

Patients with ACC are at high risk of pelvic recurrence, distant metastases, or both. The relapse rate of CC ranges between 11% and 22% for FIGO stages IB–IIA, and between 28% and 64% for FIGO stages IIB–IVA [2,5,6]. Several prognostic factors for recurrence and survival such as FIGO stage, tumour size, nodal status, or presence of lympho-vascular space involvement (LVSI) have been reported.

The concept of platinum sensitivity, defined by the period from the completion of an initial platinum-based chemotherapy to disease recurrence, is recognized as a major prognostic factor for patients with ovarian cancer [7,8,9]. However, for AAC the prognostic value of the platinum sensitivity concept has not yet been evaluated in terms of recurrence pattern, survival, independent prognostic factors, or clinical value.

Hence, based on a multi-institutional database from the FRANCOGYN Group, we set out to (i) validate the applicability of the platinum sensitivity concept, (ii) estimate its prognostic interest in terms of overall survival (OS), and (iii) stratify its value regarding the pattern of recurrence (location, timing), so as to improve recurrence management for women with FIGO 2019 stages IB3–IVA CC.

## 2. Material and Methods

### 2.1. Study Population

The data of women with FIGO 2019 stages IB3–IV histologically proven squamous cell carcinoma or adenocarcinoma, treated between May 2000 and November 2017, were retrospectively abstracted from 12 institutions from the FRANCOGYN Group with maintained CC databases. All enrolled women gave their consent for their information to be used for research purposes. The study was approved by the Ethics Committee of the National College of French Gynaecologists and Obstetricians (CNGOF) (CEROG 2016-GYN-0502).

All enrolled women underwent preoperative workup including history, physical examination, cervical biopsy, magnetic resonance imaging (MRI) +/− positron emission tomography–computed tomography (PET-CT) if indicated, according to the FIGO stage and the period of treatment.

Clinical, surgical, pathological, and treatment data were collected: patients’ age, body mass index (BMI; calculated as weight in kilograms divided by the square of height in metres), lymph node (LN) status, clinical FIGO stage, final pathological analysis (histological type, tumour grade, LVSI, and tumour size), treatment modalities, date and type of recurrence, date of death or last contact, and cause of death. 

We reviewed all cases and re-staged them according to the FIGO 2019 classification after final pathological analysis [10].

### 2.2. Therapeutic Management

Therapeutic management was decided on by a multidisciplinary committee on an individual basis, according to the current French National Institute of Cancer (INCa) guidelines [11]. Patients were included when initially treated with platinum-based chemotherapy, either CCRT +/− vaginal brachytherapy (VBT) or CT alone.

Clinical follow-up consisted of physical examination and the use of imaging techniques according to the findings. Follow-up visits were conducted every 6 months for 5 years, and then annually.

Recurrent disease was assessed by physical examination, imaging techniques (ultrasonography, CT, MRI, PET-CT), and histological findings when feasible.

## 3. Definition and Classification of Recurrence

### 3.1. Definition of Platinum Sensitivity

According to previous studies for ovarian cancer, patients were defined as being platinum resistant (PR) if recurrence occurred <6 months after the last chemotherapy cure, partially sensitive (PPS) if between 6 and 11 months, sensitive (PS) between 12 and18 months, or very sensitive (PVS) for recurrence occurring >18 months [12].

### 3.2. Recurrence Classification

According to our previously published rTNM system [13], recurrences were classified as follows: 1/ locoregional, including recurrence in the vaginal vault alone, rT1; centropelvic recurrence with or without vaginal involvement, rT2; peritoneal carcinomatosis and or ascites, rT3; 2/ nodal, including infradiaphragmatic nodal recurrence, rN1; or supradiaphragmatic nodal recurrence, rN2; or 3/ distant organ metastasis, rM1. Multiple-site recurrence was defined as recurrence with more than one pathway of dissemination.

### 3.3. Statistical Analysis

The patients, tumours, and treatment characteristics were analysed using chi-square statistics or Fisher’s exact test for categorical variables, and the *t*-test or analysis of variance (ANOVA) for continuous variables.

Kaplan–Meier estimates were used to estimate event-time distributions, and the log-rank test was used to compare the differences among the different groups in terms of recurrence-free survival (RFS) and OS. Time to the first CC recurrence for a specific site was evaluated by using cumulative incidence analysis (Gray’s test) and competing risks regression analysis to estimate sub-distribution hazard ratios (HRs) and 95% confidence intervals (95% CIs).

Values of *p* < 0.05 were considered to denote significant differences. Data were managed with an Excel database (Microsoft, Redmond, WA, USA) and analysed using the R 2.15 software, available online.

## 4. Results

### 4.1. Population Characteristics

During the study period, 668 women with FIGO 2019 stages IB–IV CC were documented as having received treatment: 168 at Tenon University Hospital (25%), 100 at Rennes University Hospital (15%), 99 at Tours University Hospital (15%), 59 at La Pitié University Hospital (9%), 56 at Lyon South University Hospital (8%), 55 at Creteil University Hospital (8%), 41 at Strasbourg University Hospital (6%), 32 at Jeanne de Flandre University Hospital (5%), 16 at Jean Verdier Hospital (3%), 17 at Marseille North University Hospital (3%), 14 at Poissy Hospital (2%), and 11 at Reims University Hospital (1%). The patient flowchart is shown in Figure 1.

The median age of the women was 52.0 years (range 44–63 years), and their median BMI was 24 kg/m^2^ (range 21–29). Initial treatment consisted of CCRT with platinum-based chemotherapy +/− VBT or CT alone for about 65% of the patients, and the rest underwent CCRT with platinum-based chemotherapy +/− VBT followed by radical hysterectomy.

The epidemiological, therapeutic, and histological characteristics of the population are reported in Table 1.

### 4.2. Overall Survival

#### 4.2.1. Whole Population

The median follow-up was 36.1 months and the respective 3-year OS and RFS were 55.6% (95% CI, 48.4–63.9) and 68.2% (95% CI, 64.4–72.3), respectively.

#### 4.2.2. OS According to Platinum Sensitivity

The respective 3-year OSs according to the patients’ platinum status were: 14.6% (95% CI, 4.2–50.2), 26.4% (95% CI, 13.6–50.8), 22.8% (95% CI, 10.9–47.4), and 52.2% (95% CI, 40.8–66.8), (*p* < 0.001) for women in the PR, PPS, PS, and PVS categories, respectively (Figure 2). The median duration to recurrence was 4, 9, 14, and 26 months, respectively. The median survival after recurrence was 6, 15, 29, and 46 months, respectively (Figure 3).

#### 4.2.3. Stratification Analysis

Three prognostic groups were defined according to the respective 3-year OS after recurrence: a low-risk group, defined by recurrence after 18 months; an intermediate-risk group, defined by recurrence between 6 and 17 months; and a high-risk group, defined by recurrence <6 months. The OS after recurrence for these three groups is reported in Figure 4.

The respective 3-year OSs were 52% (95% CI: 40.8–66.8), 21.6% (95% CI: 12.6–37.2), and 14.6% (95% CI: 4.2–50.2), for patients in the low-, intermediate-, and high-risk groups, respectively (*p* < 0.001). The median duration to recurrence was 4, 11, and 26 months, respectively. The median survival after recurrence was 6, 21, and 46 months, respectively. 

### 4.3. Recurrence Pattern According to Platinum Sensitivity

#### 4.3.1. Recurrence

In the overall population, the recurrence rate was 29.7% (199/668), and the death rate was 22.9% (153/668). The median time to recurrence was 27.5 months (range 1–140 months), with 40.2% (80/199) of recurrences occurring within the first year. Among the recurrences, 29% (57/199) were local, 10% (20/199) nodal, and 61% (122/199) involved a distant organ.

According to platinum sensitivity stratification, recurrences were locoregional, nodal, and distant in 39%, 45%, and 41% for the low-risk group; 56%, 50%, and 46% for the intermediate-risk group; and 5%, 5%, and 13% for the high-risk group (*p* < 0.001). These data are summarized in Table 2.

#### 4.3.2. Cumulative Recurrence Curves

The patterns of recurrence and cumulative curves are summarized in Table 2.

Cumulative recurrence curves for overall population and prognostic subgroups are summarized in Table 2.

### 4.4. Predictive Factors for OS in Multivariate Analysis 

In multivariate analysis, LVSI, platinum sensitivity, and LN status were prognostic factors for OS after recurrence. The results are summarized in Table 3.

## 5. Discussion

To the best of our knowledge, this is the first multi-institutional study conducted in Europe demonstrating the applicability of the platinum sensitivity concept for patients with recurrent ACC.

As previously reported for ovarian cancer, we demonstrated that platinum sensitivity status may help to classify recurrent patients in different categories. By analogy with the study of Takekuma et al., we classified patients depending on time to recurrence: <6 months, between 6 and 11 months, between 12 and 18 months, and >18 months, thus allowing the definition of three prognostic subgroups for OS after recurrence (i.e., low, intermediate, and high risk).

In addition, the platinum sensitivity status appears to be a strong prognostic factor correlated to the pattern of recurrence, with significantly higher risk of metastatic or multisite recurrence for PR patients (recurrence <6 months) and, in con, higher rates of local or isolated infradiaphragmatic nodal recurrence for PVS patients (recurrence >18 months).

We hypothesized that platinum sensitivity is of: (i) high future clinical value for improving management after recurrence by defining the best treatment option (radiotherapy, surgery, second-line chemotherapy, supportive care); (ii) great research interest to design future clinical trials, since there is currently no consensus on the standard of care for second-line systemic treatment of recurrent/metastatic CC.

Platinum sensitivity is a well-reported powerful prognostic factor for recurrent ovarian cancer [12]. However, its prognostic value in terms of recurrence pattern [11,13], survival, independent prognostic factors, and clinical implications has been poorly studied for CC [14,15,16,17]. The most relevant study in this setting was conducted by Takekuma et al. [15] in 677 patients with CC from 71 centres in Japan. They demonstrated that the median OS for patients with a platinum-free interval (PFI) of <6, 6–11, 12–17, and ≥18 months was 12.1 (95% CI, 11.0–14.1) months, 17.4 (15.5–20.4) months, 20.2 (17.9–27.6) months, and 29.9 (26.7–36.0) months, respectively (*p* < 0.0001, log-rank). Similarly, we observed that the respective 3-year OSs according to the patients’ platinum status were: 14.6% (95% CI, 4.2–50.2), 26.4% (95% CI, 13.6–50.8), 22.8% (95% CI, 10.9–47.4), and 52.2% (95% CI, 40.8–66.8), (*p* < 0.001) for women in the PR, PPS, PS, and PVS categories, respectively. This similar prognostic pattern found in both populations—Japanese and French—confirms the applicability of the concept of platinum sensitivity in CC. However, further prospective studies are needed to confirm these promising results. The apparently higher OSs in our study, especially for very sensitive (PVS) patients, could be explained by FIGO stages and treatment disparities between the two studies. In our cohort 62% of the women had FIGO stages I–II and 38% stages III–IV, versus 52% and 47% in the Takekuma et al. cohort, respectively. In addition, more women underwent chemotherapy alone in Takekuma et al.’s study associated with a higher rate of ACC. We defined three groups of recurrence risk (i.e., patients with recurrence occurring >18 months, between 6 and 17 months, and <6 months) for which the prognosis after recurrence appeared good, intermediate, and poor, respectively. In this setting, Tanioka et al. previously showed that a time to recurrence of 12 months was an independent predictive factor of tumour response, and that a 6 month PFI was an independent prognostic factor of OS for patients with recurrent CC [16]. Takekuma et al. suggested that a PFI of 12 months significantly influenced both progression-free survival and OS [14], and that the best cut-off value of PFI that affected OS was 7 months as calculated by the minimum *p*-value method [15]. In the study of Matoda et al. including women with metastatic and recurrent CC, a PFI of more than 24 months was the discriminating point between platinum-sensitive and resistant patients [17,18]. The main clinical interest of such stratification in practice is to carefully classify the recurrent population, considering the heterogeneity of both the tumour and patient characteristics. For patients at low-, intermediate-, and high-risk, the OSs were: 0.52 (0.41–0.67), 0.22 (0.13–0.37), and 0.14 (0.043–0.50) (*p* < 0.001), respectively. Our work suggests that defining risk groups according to platinum sensitivity status may help to identify clinical situations where multimodality therapy should be proposed for high-risk patients (patients with metastatic and non-operable recurrent CC) who are generally not amenable to therapy with curative intent, and for whom the goal of treatment is symptom palliation and prolongation of survival with systemic therapy. This stratification of risk groups is of particular value since the emergence of immunotherapy in ACC, especially now that the causative role of the human papillomavirus (HPV) is well established. A number of immunotherapy trials have been conducted to evaluate vaccine-based therapies, adoptive T-cell therapy, and immune-modulating agents, for example with PD-1 and PD-L1 blockade. Additionally, these treatments are still in the investigation phase, and they have shown promising results for progressing or recurrent patients [19].

Patients with CC may develop pelvic, infra-, and/or supradiaphragmatic nodal metastases, distant metastases, or a combination of both. The recurrence rate varies significantly: from 22%–55% for locoregional recurrence, to 22%–75% for distant recurrence, and 2%–50% for combined recurrence [2,4,5]. In this setting, we recently reported the first standardized and reproducible classification of CC recurrence based on prognosis and anatomical distribution [13]. In our population, the overall recurrence rate was 29.7% (199/668), with 40.2% (80/199) of recurrences occurring during the first year. Among the recurrences, 29% (57/199), 10% (20/199), and 61% (122/199) were rT, rN, and rM, respectively. Similar results were reported by Takekuma et al., with 34.1%, 42.7%, and 23.2% for pelvic, distant, and both localisations, respectively. Additionally, we demonstrated that the pattern (location) of recurrence is also influenced by the platinum sensitivity status. Indeed, recurrences were most likely to be local for women in the low-risk group (recurrence >18 months, i.e., PVS patients), and metastatic or multisite for women in the high-risk group (recurrence <6 months, i.e., PR patients). More precisely, recurrence was locoregional, nodal, and distant in 39% (22), 45% (9), and 41% (50) for the low-risk group; 56% (32), 50% (10), and 46% (56) for the intermediate-risk group; and 5% (3), 5% (1), and 13% (16) for the high-risk group (*p* < 0.001), respectively. This is of particular interest for improving post-treatment monitoring programmes, which currently differ widely from country to country.

The management of women with recurrent CC remains a major issue in gynaecologic oncology. However, as underlined by the ESMO Clinical Practice Guidelines for CC, there is a major knowledge gap on this topic due to the lack of well-designed studies [2,5,20]. Schematically, two situations can be distinguished: women for whom surgical resection or radiotherapy may potentially be curative for locally recurrent or limited metastatic disease, but often not feasible (representing as many as 38.7% (77/199) of the present cohort); and women with recurrent or metastatic disease not amenable to therapy with curative intent, who are usually administered a systemic anti-cancer therapy with palliative intent. However, there is currently no consensus on the standard of care for second-line systemic treatment of recurrent/metastatic cervical cancer. Some randomized trials have reported a response rate (RR) of 31%–36% for a platinum-based combination treatment [21,22,23,24]. Tewari and Monk reported the impact of prior platinum exposure in this setting. The RR in platinum-naive patients was 20% for cisplatin monotherapy, 39% for topotecan CT, and 37% for a paclitaxel combination CT. The respective RR after previous cisplatin-based exposure was estimated at 5%–8% for cisplatin monotherapy, 15% for the combination of cisplatin with topotecan, and 32% for paclitaxel, respectively [24]. This suggests that a history of platinum exposure could be an indicator in predicting the response of second-line systemic treatment. In agreement with these results, we also reported that platinum status is one of the most powerful prognostic factors for OS after recurrence, along with the LVSI and nodal status. Indeed, we observed a hierarchical HR between platinum-sensitive and resistant cases (*p* < 0.001). The prognostic impact of platinum status, and the current absence of an effective therapeutic alternative, underline the need for a better understanding of the chemoresistance mechanism in CC patients. Several studies have focused on various molecular and genetic factors such as microRNAs [25], tanshinone I [26], enhancer of zeste homolog 2 [27], and polymorphisms in the PI3K/Akt pathway [28] among others, with promising results.

The strengths of our study lie in its multicentre nature and the large number of women included, but some limitations deserve to be mentioned. First, there is an inherent bias due to its retrospective nature. Indeed, the guidelines changed during the period of data collection, in particular with modification of the FIGO staging system and introduction of PET-CT. This incurred modifications to the management strategy, especially concerning LN staging, and accuracy in the diagnosis of recurrence. However, all included women were treated in regional referral centres applying French/European guidelines after systematic multidisciplinary committee approval. Second, follow-up modalities were highly heterogeneous, which could have impacted the reported time to recurrence. Third, results should be analysed with caution since some authors suggest that prior radio-sensitization could impact the effectiveness of second-line chemotherapy, in particular via radiation effects on bone marrow function or by hindering drug distribution in previously irradiated tissues [29,30]. The lack of precise data concerning radiotherapy is thus also a limit. Finally, although we reported that platinum status emerged as one of the most powerful prognostic factors for OS, the lack of precise information concerning the treatment of recurrence may impact the interpretability of such results.

## 6. Conclusions

In conclusion, our study confirms the applicability of the platinum sensitivity concept in women with ACC, with a strong impact on survival after recurrence among patients previously treated with platinum chemotherapy. As most recurrences were not amenable to therapy with curative intent, a history of platinum exposure could be a major indicator to predict second-line systemic treatment response. Although screening and vaccination undeniably represent the cornerstone to relieve the global burden of CC, further prospective studies should be conducted to better understand the chemoresistance mechanisms of ACC.

## Figures and Tables

**Figure 1 jcm-09-03646-f001:**
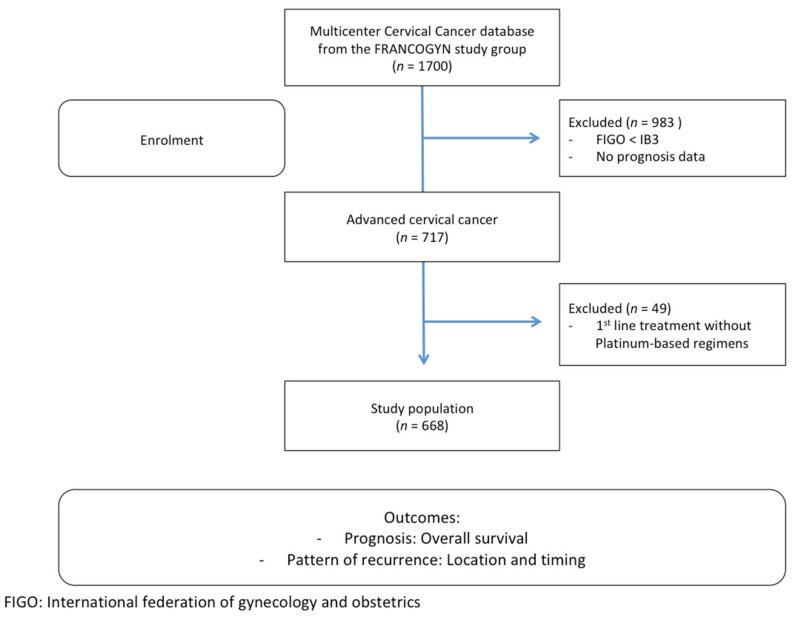
Patients flowchart.

**Figure 2 jcm-09-03646-f002:**
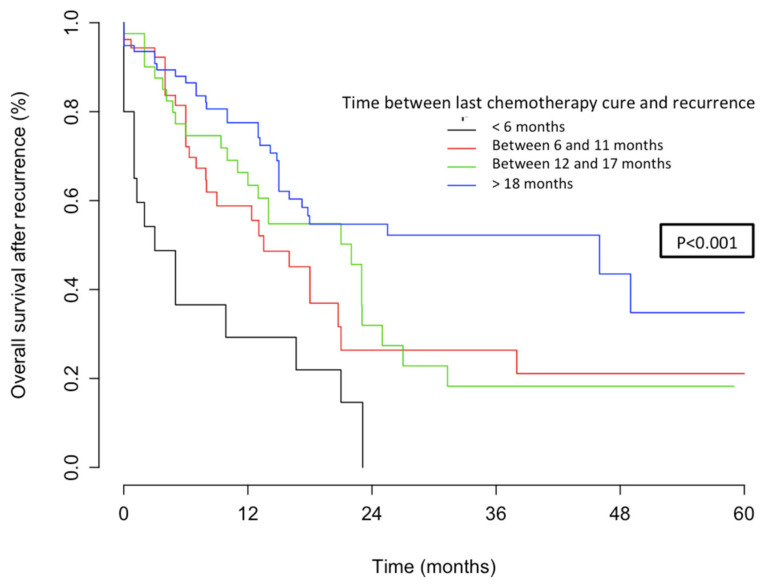
Overall survival after recurrence according to platinum sensitivity.

**Figure 3 jcm-09-03646-f003:**
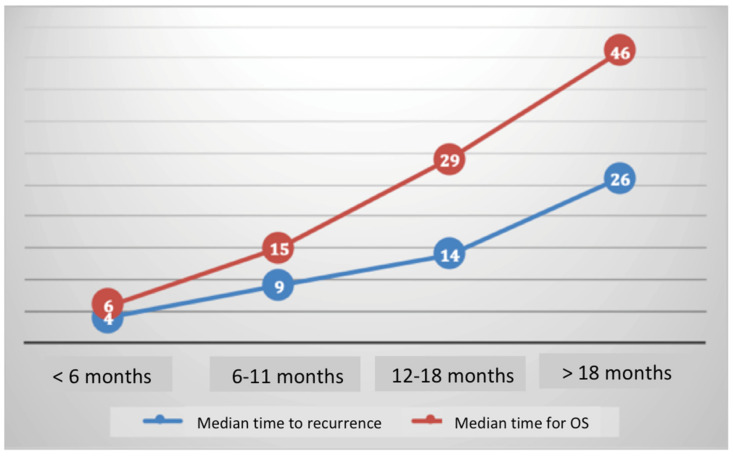
Median time to recurrence and overall survival according to platinum sensitivity.

**Figure 4 jcm-09-03646-f004:**
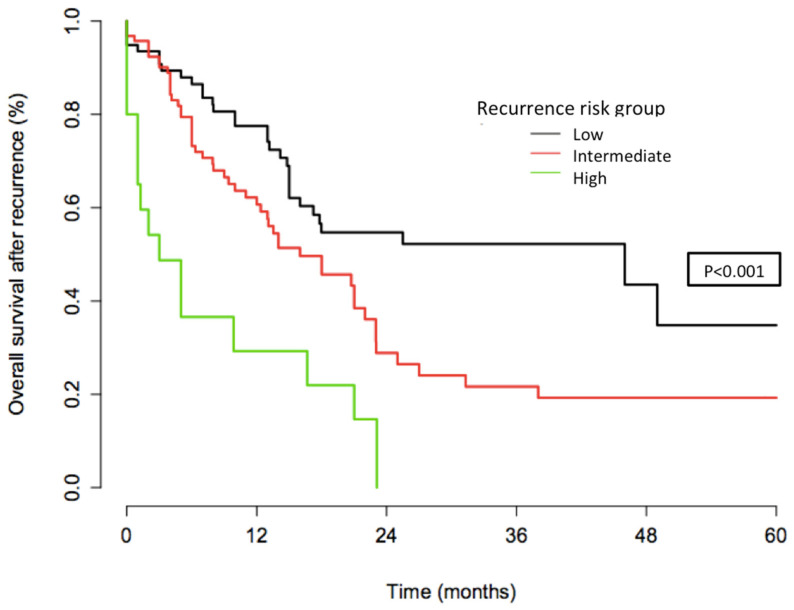
Overall survival after recurrence according to recurrence risk group.

**Table 1 jcm-09-03646-t001:** Epidemiological and histological characteristics of the whole population.

Features and Characteristics	Total Population (*N* = 668)
**Age** at diagnosis, median (IQ)	52 (44−63)
**BMI** (kg/m^2^), median (IQ)	24 (21–29)
**Tumour size** (mm), median (IQ)	30 (24–35)
**FIGO stage**, *N* (%)	
- IB3	44 (7%)
- IIA	53 (8%)
- IIB	318 (47%)
- IIIA or B	56 (8%)
- IIIC	113 (17%)
- IVA or B	84 (13%)
**Histology**, *N* (%)	
- Squamous cell carcinoma	585 (88%)
- Adenocarcinoma	83 (12%)
**Histological grade**, *N* (%)	
- 1	176 (27%)
- 2	187 (28%)
- 3	140 (21%)
- Unknown	165 (24%)
**Lympho-vascular space invasion**, *N* (%)	
- Positive	63 (9%)
- Negative	225 (34%)
**Treatments**, *N* (%)	
- CCRT +/− VBT followed by RHT	211 (35%)
- CCRT+/− VBT or CT alone	457 (65%)
**Final lymph node status**, *N* (%)	
- Positive	113 (17%)
- Negative	244 (36%)
- Unknown/no surgical staging *	311 (47%)
**Platinum sensitivity**, *N* (%)	
- No recurrence	469 (70%)
- Platinum resistant	20 (3%)
- Platinum partially sensitive	55 (8%)
- Platinum sensitive	43 (6%)
- Platinum very sensitive	81 (12%)
**Recurrence location**, *N* (%)	
- Locoregional (rT)	57 (9%)
- Nodal (rN)	20 (3%)
- Distant organ (rM) and multisite	122 (18%)

Abbreviations: IQ: interquartile; BMI: body mass index; FIGO: international federation of gynaecology; CCRT: concomitant chemoradiotherapy; CT: chemotherapy; RHT: radical hysterectomy; VBT: vaginal brachytherapy. * because there is no staging or no data on staging.

**Table 2 jcm-09-03646-t002:** Recurrence pattern and cumulative recurrence curves according to platinum sensitivity.

Population (*N*)	Recurrence Site (*N*)	3 Years Cumulative Rates	*p*-Value	Figure
**Overall** **Population** (668)	- rT (57) - rN (20) - rM or multisite (122)	41.9% (25.2–54.9) 28.4% (8.1–44.1) 49.8% (38.4–59)	*p* < 0.001	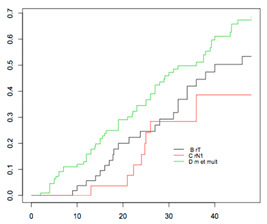
**Low-risk** **Group** (81)	- rT (22) - rN (9) - rM or multisite (50)	35.8% (10.2–54.1) 12.5% (0–32.7) 61.2% (40.1–74.9)	*p* = 0.057	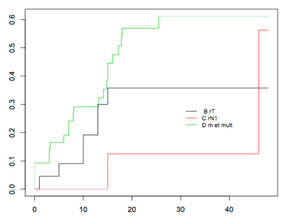
**Intermediate-risk group** **(98)**	- rT (32) - rN (10) - rM or multisite (56)	66.2% (29.8–83.7) 70.8% (13.2–90) 86.8% (64.2–95.1)	*p* = 0.1701	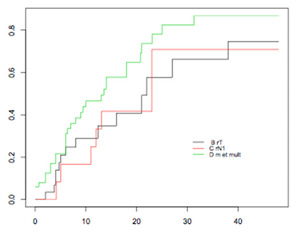
**High-risk** **Group** **(20)**	- rT (3) - rN (1) - rM or multisite (16)	75% (0–95.4) 77.8% (35–92.4)	*p* = 0.4522	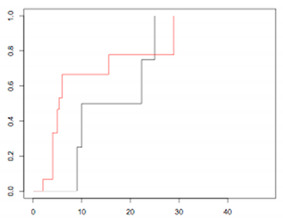

**Table 3 jcm-09-03646-t003:** Multivariate analyses of prognostic factors regarding OS after recurrence.

	HR	95% CI	*p*
**Histology**, *N* (%)			
- Squamous cell carcinoma	1		
- Adenocarcinoma	1.92	0.64–5.69	0.23
**Histological grade**, *N* (%)			
- 1	1		
- 2	0.71	0.29–1.70	
- 3	1.99	0.92–4.33	0.079
**LVSI**, *N* (%)			
- Positive	1		
- Negative	0.39	0.20–0.76	0.005
**Final lymph node status**, *N* (%)			
- Unknown*/no surgical staging	1		
- Positive	0.48	0.19–1.18	
- Negative	0.21	0.08–0.53	0.001
**Treatments**, *N* (%)			
- CRT+/− VBT or CT alone	1		
- CCRT+/− VBT followed by RHT	0.74	0.34–1.60	0.459
**Platinum status**			
- Very sensitive	3.08	1.36–6.96	
- Sensitive	7.87	3.15–19.65	
- Partially sensitive	19.0	5.85–61.67	
- Resistant	17.08	2.94–98.99	<0.0001

Abbreviations: LVSI: lympho-vascular space invasion; CCRT: concomitant chemoradiotherapy; CT: chemotherapy; VBT: vaginal brachytherapy; RHT: radical hysterectomy.

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
