# Peer review of "Recurrence Pattern of Cervical Cancer Based on the Platinum Sensitivity Concept: A Multi-Institutional Study from the FRANCOGYN Group"

_jcm, 2020, doi:10.3390/jcm9113646_

Round 1

Reviewer 1 Report

The aim of the paper was to validate the usefulness of the platinum-sensitivity concept (known from ovarian cancer) in prognosis of advanced cervical cancer (ACC), as well as, to stratify its value regarding the pattern of recurrence. Authors conclude that platinum-sensitivity status may help to classify patients with ACC into 3 prognostic groups for OS after recurrence and appears to be a prognostic factor for the pattern (local, metastatic) of recurrence.

Comments:

1/ Table 1 rows “Treatments” and the next one – radio-chemotherapy was followed by radical hysterectomy(RHT) in 35% of patients. But “Final lymph node status” (next row in the table) presents data from 17% (positive) + 36% (negative) = 53% of patients. So 18% of studied women had lymph node staging performed not during radical hysterectomy? Authors should indicate clearly the way of staging in this group of patients.

2/ The patients had either radio-chemotherapy followed by RHT or radio-chemotherapy/ chemotherapy alone (Table 1). In my opinion the concept of platinum-sensitivity as a prognostic factor for ACC patients should be additionally recalculated separately for group having RHT and not having RHT, as radical hysterectomy could bias the results. Authors should prove to the readers that it was not the case.  

Author Response

Dear Editors and Reviewers,

We would first like to thank the reviewers for their constructive comments. All remarks were addressed and modified accordingly.

Here are our detailed modifications:

Reviewer #1:

The aim of the paper was to validate the usefulness of the platinum-sensitivity concept (known from ovarian cancer) in prognosis of advanced cervical cancer (ACC), as well as, to stratify its value regarding the pattern of recurrence. Authors conclude that platinum-sensitivity status may help to classify patients with ACC into 3 prognostic groups for OS after recurrence and appears to be a prognostic factor for the pattern (local, metastatic) of recurrence.

Comments:

1/ Table 1 rows “Treatments” and the next one – radio-chemotherapy was followed by radical hysterectomy (RHT) in 35% of patients. But “Final lymph node status” (next row in the table) presents data from 17% (positive) + 36% (negative) = 53% of patients. So 18% of studied women had lymph node staging performed not during radical hysterectomy? Authors should indicate clearly the way of staging in this group of patients.

We take into account this very relevant comment. As requested in French guidelines, these 18% have been staged by exclusive lombo-aortic lymphadenectomy, in order to estimate the rate of positive lymph nodes in lombo-aortic area to optimize radiotherapy fields.

2/ The patients had either radio-chemotherapy followed by RHT or radio-chemotherapy/chemotherapy alone (Table 1). In my opinion the concept of platinum-sensitivity as a prognostic factor for ACC patients should be additionally recalculated separately for group having RHT and not having RHT, as radical hysterectomy could bias the results. Authors should prove to the readers that it was not the case.  

Thank you for this comment. As previously published by several authors (Morice P, Rouanet P, Rey A, et al. Results of the GYNECO 02 study, an FNCLCC phase III trial comparing hysterectomy with no hysterectomy in patients with a (clinical and radiological) complete response after chemoradiation therapy for stage IB2 or II cervical cancer. The Oncologist. 2012;17(1):64–71 or Lèguevaque P, Motton S, Delannes M, Querleu D, et al. Completion surgery or not after concurrent chemoradiotherapy for locally advanced cervical cancer? Eur J Obstet Gynecol Reprod Biol. 2011 Apr;155(2):188–92, for example) the impact of RHT has never been clearly demonstrated in the management of patients with locally advancerd cervical cancer. Hence, we considered this sub analysis was not relevant to (in)validate the concept of platinum-sensitivity.

Reviewer 2 Report

The manuscript titled “Recurrence pattern of cervical cancer based on the platinum sensitivity concept: a multi‐institutional study from the FRANCOGYN Group” reports an interesting and fairly new multi-institutional retrospective study on a sample of 668 French women (similar to what Munetaka Takekuma did in 2017 on a sample of 677 Japanese women) that evaluate whether platinum sensitivity could be identified as a significant prognostic factor for women with recurrent cervical cancer who have had previous treatment with platinum chemotherapy.

Overall, I feel this manuscript is of good quality, written quite clearly and appropriate for Journal of Clinical Medicine. In my opinion, the manuscript is written in good English (but I am not a native speaker, unfortunately). This manuscript is basically publishable as is.

However, only one minor issue need to be addressed in order to improve the quality of the paper about obviousness, as discussed below:

In table 1, the data of the lines “Platinum resistant”, “Platinum partly sensitive”, “Platinum sensitive” and “Platinum very sensitive” are inverted with respect to the indication “N (%)”.

Author Response

Dear Editors and Reviewers,

We would first like to thank the reviewers for their constructive comments. All remarks were addressed and modified accordingly.

Here are our detailed modifications:

Reviewer #2:

The manuscript titled “Recurrence pattern of cervical cancer based on the platinum sensitivity concept: a multiinstitutional study from the FRANCOGYN Group” reports an interesting and fairly new multi-institutional retrospective study on a sample of 668 French women (similar to what Munetaka Takekuma did in 2017 on a sample of 677 Japanese women) that evaluate whether platinum sensitivity could be identified as a significant prognostic factor for women with recurrent cervical cancer who have had previous treatment with platinum chemotherapy.

Overall, I feel this manuscript is of good quality, written quite clearly and appropriate for Journal of Clinical Medicine. In my opinion, the manuscript is written in good English (but I am not a native speaker, unfortunately). This manuscript is basically publishable as is.

Thank you for this review.

However, only one minor issue need to be addressed in order to improve the quality of the paper about obviousness, as discussed below:

In table 1, the data of the lines “Platinum resistant”, “Platinum partly sensitive”, “Platinum sensitive” and “Platinum very sensitive” are inverted with respect to the indication “N (%)”.

We considered this comment and added the correction in the manuscript.

Reviewer 3 Report

I would like to congratulate Dr Foucher and colleagues for the great collaborative effort and for mantaining such a large database accross different French centres.

The results of their research is intriguing, and I have included immediately below a few items for the authors to consider addressing.  

-Abstract Methods (Pg 1) and Materials and Methods (Pg 3): Given that all patients included in the study received treatment before the updated FIGO 2019 staging, can the authors clarify if the staging was reviewed case-by-case and documented as per FIGO 2019 staging?

-Abstract 51-52 and table 3: “In multivariate analysis, platinum sensitivity status…” By what was the multivariate analysis adjusted (i.e. stage…)? Can you clarify if this was a univariate or multivariate analysis?

-Abstract interpretation (53-55), please rephrase. Doesn’t read easily and the fact that it is a prognostic factor is repeated twice in the sentence.

-Please correct reference 1. Indicate: International Agency for Research on Cancer. World Health Organization… Please also add the date where the website was accessed for the information in the reference.

-Introduction (69-70): I think its important to differentiate the locally-advanced vs the truly metastatic (IVB) cervical cancers, given its therapeutic and prognostic implications. Consider including some information regarding that instead of merging them all in the advanced category. 

-Introduction (73): Consider adding also information about Bevacizumab (either in the introduction or conclusion). The trial is cited later, but there is no mention on antiangiogenics in the manuscript.

-Materials and methods (91): Please clarify if all stage IV or only stg IVA were eligible.

Please clarify if there was any limitation in terms of histology because I only see squamous and adenocarcinomas included. Where adenosquamous and other rare histologies excluded?

-Materials Methods (110-111): Is the fact that the case was discussed in multidisciplinary committee recorded as part of the dataset? As per the statement it seems that all the cases were discussed.

-Page 3. Platinum sensitivity: Consider using the term: "partially platinum-sensitive" instead of partly across the text. This is the term more frequently used in ovarian cancer. 

I believe that the reference that you used does not mention the term “very sensitive” for over 18 months, and this is not usually used in standard practice or for clinical trial inclusion/exclusion in ovarian cancer. Please update with an appropriate reference.

Additionally, in ovarian cancer the term “platinum-refractory” is often used to define patients that progress during platinum or within one month from the last platinum therapy. Did you find any platinum refractory patients in your cohort? Was there any correlation?

-Statistical analysis (132): “The women, tumours and treatment characteristics..” Consider using the term patients baseline characteristics instead of “women”.

-Results. Patients: (145) Clarify the stage IVA or all stage IV (As per introduction IVA, but not clear).

Results patients. Figure 1: As per the figure it appears that the only platinum regimen used was cisplatin. Where patients that received Carboplatin excluded (i.e. renal disfunction)? Please clarify also this in the methods section.

- Table 1. Was there any data of ethnicity collected? Authors argue that this study adds to the Japanese study given that it is done in European population in the conclusions. Although we can assume that the population may be predominantly white given that data is from French institutions, If available, it would be nice to see the ethnicity of the patients included. There is growing evidence that this can have in prognosis impact in certain gyne cancers.  

-4.2.2. 172 and Figure 1. There is a cross-over of the Kaplan-meier curves at least 3 times, so authors should be careful when stating the significance and p value of the results. Authors may also want to consider providing information about Plat Sensitive (over 6m) vs Plat Resistant, the K-M curve separation may be more clear with the two variables.

-4.2.2. Figure 3: Specify what the numbers (4,6,…) are in the figure legend. Does it correspond to months?

-Figure 4: There is no legend

-Table 2. Could authors add proportion and number of pts per site of recurrence in the table?

-Table 3. Platinum sensitivity. Authors may consider adding also sensitive vs resistant as two variables if appropriate.

 -Discussion (213-216): As indicated earlier, the 18 months cut-off is not usually used in ovarian cancer. Maybe authors should mention the Japanese cervical cancer study to justify this cut-off time. Please add a reference to the paragraph.

-Discussion (223): Consider using the term “best supportive care” instead of “best support”.

-Discussion (238): “…confirms the applicability of the “concept” of platinum sensitivity in CC.” I suggest the authors to be careful with this statement. This is retrospective data in a highly prevalent tumour site. I think that a confirmation and proper validation needs to be prospective.

-(261-263): “This stratification of risk groups is of particular  value since the emergence of immunotherapy in ACC, especially now that the causative role of the human papillomavirus (HPV) is well established”. To my knowledge HPV status is not yet used as a biomarker of response of ICI in cervical cancer. I think that PDL1 status has been better studied in this setting but still not good data. It is also not clear to me how the stratification would help to guide who would benefit from immunotherapy.

-(263-265): “…number of immunotherapy trials have been conducted to evaluate vaccine-based therapies, adoptive T-cell therapy and immune-modulating agents”. The reference that you are using is from 2015, there have been several updates in the immune-oncology setting since then. Instead of mentioning that a number of trials have been conducted, consider talking about the results of the actual studies and mention results (several studies with Pembro and Nivo with results already reported; KEYNOTE-158 (NCT02628067) CheckMate358…).

-284-286: The authors mention the ESMO clinical guidelines. However, none of the references (2)(5)(7) refer to the guidelines. Please update. Additionally, reference 7 does not fit there as it is an ovarian cancer study.

-(284-323): Please mention the study by Tewari et al assessing treatment with bevacizumab along with platinum based chemotherapy (your reference 23). Bevacizumab has been so far the only targeted therapy demonstrating an improvement in overall survival in this setting. Additionally, I think that the reference is not listed in the correct place (297-299: response post-platinum).

-(303): “refractory cases”. Do the authors mean refractory (pg during treatment or within 1 month from the last platinum) or resistant (<6m).

-(306-308): “Several studies have focused on various molecular and genetic factors such as 306 MicroRNAs (24), tanshinone I (25), enhancer of zeste homolog 2 (26), and polymorphisms in 307 the PI3K/Akt pathway (27) among others, with promising results.” What do you mean by molecular and genetic factors? Biomarkers of response or treatments? Some of these examples are pre-clinical cell lines studies and other performed in patients consider being more specific. Authors may also consider to discuss ongoing clinical trials in recurrent cervical cancer.  

Author Response

Dear Editors and Reviewers,

We would first like to thank the reviewers for their constructive comments. All remarks were addressed and modified accordingly.

Here are our detailed modifications:

Reviewer #3:

I would like to congratulate Dr Foucher and colleagues for the great collaborative effort and for mantaining such a large database accross different French centres.

The results of their research is intriguing, and I have included immediately below a few items for the authors to consider addressing.  

We would like to thank Reviewer #3 for his detailed work, which has been very helpful to ameliorate our manuscript.

- Abstract Methods (Pg 1) and Materials and Methods (Pg 3): Given that all patients included in the study received treatment before the updated FIGO 2019 staging, can the authors clarify if the staging was reviewed case-by-case and documented as per FIGO 2019 staging?

We take into account this very relevant comment. We reviewed case-by-case and re-classified the patients according to the FIGO 2019 classification. To improve the manuscript, we corrected the Material and Methods section with the following sentence: “We reviewed all cases and re-staged them according to the FIGO 2019 classification after final pathological analysis ”

- Abstract 51-52 and table 3: “In multivariate analysis, platinum sensitivity status…” By what was the multivariate analysis adjusted (i.e. stage…)? Can you clarify if this was a univariate or multivariate analysis?

We performed a multivariate analysis allowing the statistical assessment of platinum sensitivity status according to several criteria: histology, histological grade, LVSI, final lymph node status, treatments and platinum status.We clarified this point in the abstract and added the following sentence to the abstract: “In multivariate analysis, platinum sensitivity status was a strong prognostic factor for OS after recurrence, independently from histological grade, lympho-vascular space involvement, final lymph node status and treatments.”

-Abstract interpretation (53-55), please rephrase. Doesn’t read easily and the fact that it is a prognostic factor is repeated twice in the sentence.

We clarified this point in the abstract and added the following sentence: “Platinum-sensitivity status may help to classify patients in three prognostic subgroups for OS after recurrence, and appears to be a strong prognostic factor correlated to the pattern of recurrence.”

-Please correct reference 1. Indicate: International Agency for Research on Cancer. World Health Organization… Please also add the date where the website was accessed for the information in the reference.

We corrected reference 1 and added the accessed date: “International Agency for Research on Cancer. World Health Organization Cancer today http://gco.iarc.fr/today/home, accessed on the 30th of october 2020.”

-Introduction (69-70): I think it’s important to differentiate the locally-advanced vs the truly metastatic (IVB) cervical cancers, given its therapeutic and prognostic implications. Consider including some information regarding that instead of merging them all in the advanced category. 

Thank you for this comment. We made this clear by adding the following sentence in the introduction: “It includes locally advanced CC (LACC – IB3 to IVA) and initially metastatic CC (IVB).”

-Introduction (73): Consider adding also information about Bevacizumab (either in the introduction or conclusion). The trial is cited later, but there is no mention on antiangiogenics in the manuscript.

We take into account this very relevant comment but this point is out of the scope of this study.

-Materials and methods (91): Please clarify if all stage IV or only stg IVA were eligible.

All stage IV patients were included. This point was underlined in the MM section, where we added the following sentence: “It includes locally advanced CC (LACC – IB3 to IVA) and initially metastatic CC (IVB).”

- Please clarify if there was any limitation in terms of histology because I only see squamous and adenocarcinomas included. Where adenosquamous and other rare histologies excluded?

To be homogeneous in term of management, we only included SCC and adenocarcinoma in the following study,. We thus clarified this point in the MM section by adding the following sentence: “The data of women with FIGO 2019 stages IB3-IV histologically proven squamous cell carcinoma or adenocarcinoma treated between May 2000 and November 2017 were retrospectively abstracted from 12 institutions from the FRANCOGYN Group with maintained CC databases.  “

-Materials Methods (110-111): Is the fact that the case was discussed in multidisciplinary committee recorded as part of the dataset? As per the statement it seems that all the cases were discussed.

As all cases included in the study provided from expert centers, they have been discussed in multidisciplinary committee (According to French guidelines).

-Page 3. Platinum sensitivity: Consider using the term: "partially platinum-sensitive" instead of partly across the text. This is the term more frequently used in ovarian cancer. 

We take into account this very relevant comment, and modified this point in the manuscript.

- I believe that the reference that you used does not mention the term “very sensitive” for over 18 months, and this is not usually used in standard practice or for clinical trial inclusion/exclusion in ovarian cancer. Please update with an appropriate reference.

This time cut-off is used by Takekuma et al in their multi institutional retrospective study on recurrent cervical cancer. By analogy with their work, we choose to apply it on our population.

Additionally, in ovarian cancer the term “platinum-refractory” is often used to define patients that progress during platinum or within one month from the last platinum therapy. Did you find any platinum refractory patients in your cohort? Was there any correlation?

In our data set, we could not distinguished between patients who progressed during platinum or within one month from the last platinum therapy and those who relapsed during the first 6 months of follow up. We thus considered them all as platinum resistant patients. However, this is a very relevant comment which could be studied in the future. Thank you for the idea.

-Statistical analysis (132): “The women, tumours and treatment characteristics..” Consider using the term patients baseline characteristics instead of “women”.

We take into account this comment and corrected this point

-Results. Patients: (145) Clarify the stage IVA or all stage IV (As per introduction IVA, but not clear).

We clarified this point.

- Results patients. Figure 1: As per the figure it appears that the only platinum regimen used was cisplatin. Where patients that received Carboplatin excluded (i.e. renal disfunction)? Please clarify also this in the methods section.

In our multicentric expert center database, it was not possible to precisely identify platinum regimen for all patient. In practice, all included patient were treated with either kind of platinum treatment. Hence, we corrected this point in the flowchart.

- Table 1. Was there any data of ethnicity collected? Authors argue that this study adds to the Japanese study given that it is done in European population in the conclusions. Although we can assume that the population may be predominantly white given that data is from French institutions, If available, it would be nice to see the ethnicity of the patients included. There is growing evidence that this can have in prognosis impact in certain gyne cancers.

In France, ethnical statistics are forbidden in clinical research. As a consequence, we can’t add any information about this important comment.

-4.2.2. 172 and Figure 1. There is a cross-over of the Kaplan-meier curves at least 3 times, so authors should be careful when stating the significance and p value of the results. Authors may also want to consider providing information about Plat Sensitive (over 6m) vs Plat Resistant, the K-M curve separation may be more clear with the two variables.

Thank you for this comment. These overall curves have been reported to be descriptive, in order to move further on analysis performed by Takekuma et al. After this 1st descriptive step, we added new information based on Figure 2 that help overcoming this cross over. The value of our recurrence risk group classification is to highlight the impact of platinum status on patient prognosis.

-4.2.2. Figure 3: Specify what the numbers (4,6,…) are in the figure legend. Does it correspond to months?

Yes it does. To clarify this point, we add a legend in the bottom of the Figure 3.

-Figure 4: There is no legend

We corrected this point

-Table 2. Could authors add proportion and number of pts per site of recurrence in the table?

We added it.

-Table 3. Platinum sensitivity. Authors may consider adding also sensitive vs resistant as two variables if appropriate.

We understand the comment. However, we chose to jibe with our 4 levels classification, 1/ to be homogeneous with previous analysis and 2/ as there is a significant hierachy between 4 levels.

 -Discussion (213-216): As indicated earlier, the 18 months cut-off is not usually used in ovarian cancer. Maybe authors should mention the Japanese cervical cancer study to justify this cut-off time. Please add a reference to the paragraph.

We thank the author for this comment and added the following sentence in the discussion paragraph: “As previously reported for ovarian cancer, we demonstrated that platinum-sensitivity status may help to classify recurrent patients in different categories. By analogy with Takekuma et al. study (15), we classified patients depending on time to recurrence: <6 months, between 6-11 months, between 12-18 months, and >18 months, thus allowing the definition of three prognostic subgroups for OS after recurrence (i.e., low-, intermediate- and high-risk).  

-Discussion (223): Consider using the term “best supportive care” instead of “best support”.

Thank you for this comment, we made the correction.

-Discussion (238): “…confirms the applicability of the “concept” of platinum sensitivity in CC.” I suggest the authors to be careful with this statement. This is retrospective data in a highly prevalent tumour site. I think that a confirmation and proper validation needs to be prospective.

We thus added the following sentence to the discussion: “However, further prospective studies are needed to confirm these promising results.”

-(261-263): “This stratification of risk groups is of particular value since the emergence of immunotherapy in ACC, especially now that the causative role of the human papillomavirus (HPV) is well established”. To my knowledge HPV status is not yet used as a biomarker of response of ICI in cervical cancer. I think that PDL1 status has been better studied in this setting but still not good data. It is also not clear to me how the stratification would help to guide who would benefit from immunotherapy.

Thank you for your comment. We hypothezised that an easily reproductive and relevant chemosensiibility classification of these patients may help to design futur propective study to assess the impact of immunotherapy in specific risk subgroups.

-(263-265): “…number of immunotherapy trials have been conducted to evaluate vaccine-based therapies, adoptive T-cell therapy and immune-modulating agents”. The reference that you are using is from 2015, there have been several updates in the immune-oncology setting since then. Instead of mentioning that a number of trials have been conducted, consider talking about the results of the actual studies and mention results (several studies with Pembro and Nivo with results already reported; KEYNOTE-158 (NCT02628067) CheckMate358…).

Thank you for the suggestion. We corrected the paragraph by adding the following sentence: “A number of immunotherapy trials have been conducted to evaluate vaccine-based therapies, adoptive T-cell therapy and immune-modulating agents, for example with PD-1 and PD-L1 blockade. Also these treatments are still in an investigation phase, they showed promising results for progressing or recurrent patients” and the following reference: “Jiménez-Lima R, Arango-Bravo E, Galicia-Carmona T, Lino-Silva LS, Trejo-Durán GE, Alvarado-Silva C, et al. IMMUNOTHERAPY TREATMENT AGAINST CERVICAL CANCER. Rev Investig Clin Organo Hosp Enfermedades Nutr. 2020”

-284-286: The authors mention the ESMO clinical guidelines. However, none of the references (2)(5)(7) refer to the guidelines. Please update. Additionally, reference 7 does not fit there as it is an ovarian cancer study.

We corrected this mistake and added the following reference: “Ballester M, Bendifallah S, Daraï E. [European guidelines (ESMO-ESGO-ESTRO consensus conference) for the management of endometrial cancer]. Bull Cancer (Paris). déc 2017”

-(303): “refractory cases”. Do the authors mean refractory (pg during treatment or within 1 month from the last platinum) or resistant (<6m).

As previously mentioned, we were not able to differentiate refractory and resistant cases. All women progressing during treatment or within 6months from the last platinum were thus called resistant. We corrected the sentence in the manuscript.

-(306-308): “Several studies have focused on various molecular and genetic factors such as 306 MicroRNAs (24), tanshinone I (25), enhancer of zeste homolog 2 (26), and polymorphisms in 307 the PI3K/Akt pathway (27) among others, with promising results.” What do you mean by molecular and genetic factors? Biomarkers of response or treatments? Some of these examples are pre-clinical cell lines studies and other performed in patients consider being more specific. Authors may also consider to discuss ongoing clinical trials in recurrent cervical cancer.  

We thank again the reviewer for this relevant comment. However, the aim of this paragraph was to evocate the numerous areas for research in the field of recurrent cervical cancer. For that purpose, we touched on several different articles.